# RL-DOVS: Reinforcement Learning for Autonomous Robot Navigation in Dynamic Environments

**DOI:** 10.3390/s22103847

**Published:** 2022-05-19

**Authors:** Andrew K. Mackay, Luis Riazuelo, Luis Montano

**Affiliations:** Aragón Institute for Engineering Research (I3A), University of Zaragoza, 50009 Zaragoza, Spain; 737069@unizar.es (A.K.M.); riazuelo@unizar.es (L.R.)

**Keywords:** reinforcement learning, autonomous navigation, dynamic environments, navigation strategies

## Abstract

Autonomous navigation in dynamic environments where people move unpredictably is an essential task for service robots in real-world populated scenarios. Recent works in reinforcement learning (RL) have been applied to autonomous vehicle driving and to navigation around pedestrians. In this paper, we present a novel planner (reinforcement learning dynamic object velocity space, RL-DOVS) based on an RL technique for dynamic environments. The method explicitly considers the robot kinodynamic constraints for selecting the actions in every control period. The main contribution of our work is to use an environment model where the dynamism is represented in the robocentric velocity space as input to the learning system. The use of this dynamic information speeds the training process with respect to other techniques that learn directly either from raw sensors (vision, lidar) or from basic information about obstacle location and kinematics. We propose two approaches using RL and dynamic obstacle velocity (DOVS), RL-DOVS-A, which automatically learns the actions having the maximum utility, and RL-DOVS-D, in which the actions are selected by a human driver. Simulation results and evaluation are presented using different numbers of active agents and static and moving passive agents with random motion directions and velocities in many different scenarios. The performance of the technique is compared with other state-of-the-art techniques for solving navigation problems in environments such as ours.

## 1. Introduction

Planning safe motions for robots or vehicles in populated and highly dynamic scenarios whilst maintaining optimal or suboptimal values for several criteria is challenging. For instance, one objective for robots is achieving goals safely and quickly whilst keeping social conventions about not invading personal spaces. The unpredictability of the environment makes classical planners designed for static or quasi-static scenarios work poorly. These planners do not consider the future motion of the involved agents, so it is not possible to a priori compute safe trajectories, mainly in dense environments. The dynamics of the robots and of the other mobile obstacles must be considered for motion planning. In the case of populated spaces, the trajectories must be planned considering that people do not have the same rules as robots for avoiding obstacles. Robot planners must consider the unpredictable behaviors of the surrounding obstacles and react and select the robot motion commands to avoid potential collisions. This approach is taken in this work, which is focused on designing and training systems for random obstacle trajectories. We can find this kind of scenario in many daily activities in airports, train stations, museums, thematic parks and hospitals, as some examples.

We develop, in this work, a new planner for dynamic environments. It is based on a previous work developed in this field [1] that developed a model, the dynamic object velocity space (DOVS), in which the kinematic and dynamic constraints of the robot and the moving obstacles, and their relative locations and velocities are modeled. Several strategies have been developed on this model to safely maneuver in this kind of scenario whilst achieving the objective of reaching the goals in near minimum time. We name the method S-DOVS. A brief explanation of this technique is summarized in Section 3.

The contribution of this paper is a new planner based on reinforcement learning (RL) dynamic obstacle velocity (DOVS) techniques (RL-DOVS). Instead of directly using the information of the sensors (vision, lidar) as inputs to the learning system (end-to-end system), we propose to profit from the dynamic information modeled in the DOVS. This model includes in the robocentric velocity space rich information about the dynamism of the environment for planning the next actions. This pre-processed information is the input to the learning system, speeding the training process with respect to other techniques that learn directly from the raw sensor information.

Two Q-learning approaches [2] have been developed: (i) RL-DOVS-A, in which the system learns the actions having the maximum utility on its own, and (ii) RL-DOVS-D, in which the system learns from a driver who selects the best action in every control period. The results of applying both methods are compared with each other and with those obtained using the original strategy-based DOVS (S-DOVS) method using simulations in different scenarios with different numbers of moving agents. A comparison with results from other state-of-the-art works is also shown.

Related work is presented in Section 2. The model for representing the environment dynamism used as an input for the learning method is summarized in Section 3. Section 4 describes an overview of the whole system. The RL-DOVS method addressed in this work is developed in Section 5. In Section 6, the experiments, and the evaluation and comparison among the three RL-based techniques developed are shown; a comparison with other state-of-the-art techniques is discussed in Section 7. Finally, some conclusions and future work are explained in Section 8.

## 2. Related Work

Many RL techniques developed in recent years have been applied to decision making in robotic tasks. In the field of autonomous navigation in dynamic environments, which is the focus of this paper, many of these techniques have been oriented to autonomous vehicle driving, following lanes or navigating around pedestrians. In [3], the authors presented an RL approach using deep Q-networks to steer a vehicle using a real-world simulation. The authors developed an end-to-end system that learns both steering the vehicle to follow a lane from one image and human driver behaviors. In [4], an adversarial inverse RL method for autonomous driving scenarios and overtaking tasks on a highway was developed. They proposed semantic rewards and actions based on human-like, high-level expressions. In [5], the authors developed a deep RL technique to learn maneuver decisions in autonomous driving, based on a semantic state that represents the most significant relationships among entities. In [6], a Curriculum Soft Actor-Critic for training a neural network policy for autonomous driving on a one-way road was addressed, achieving similar performance to that obtained with an experienced human driver. The method was applied for one ego vehicle overtaking another vehicle in the lateral lane. In [7], a deep Q-learning technique was used for controlling an autonomous vehicle. This method allows one to work with a continuous set of states and actions using neural networks. However, the learning process using this kind of technique is resource-intensive. Although the autonomous driving tasks established in these papers are somewhat different from the objectives in our work, we reference them because they use different RL techniques in scenarios with few moving obstacles or spend more learning time than required in our work.

In [8], the authors addressed a deep RL network for navigation in crowded environments by separately processing static and dynamic object information for learning motion patterns. Based on high-level obstacle information, Ref. [9] proposed a semantic deep RL navigation approach, using high-level object information representing specific behaviors to recognize dangerous areas around the obstacles to enhance safety. The system was evaluated in scenarios with up to nine obstacles. Recent works on learning techniques to learn socially cooperative policies [10,11,12] incorporated pedestrian-collision-avoidance strategies without assuming any particular behavior rules, using a deep RL technique. In [13], the authors addressed the challenge of navigating through human crowds. They proposed a map-based deep RL approach using an environmental map that includes the robot shape and observable appearances of obstacles and a pedestrian map that specifies the movements of pedestrians around the robot. Better training was obtained by applying both maps as inputs to the neural network. In [11,13], the system learns and tests the method in scenarios with a much lower number of moving obstacles than ours. In [10], the authors applied one specific scenario with a similar number of obstacles to ours. These works are evaluated in scenarios that are simpler and have fewer moving obstacles than ours; therefore, they are not further considered for comparison in this paper.

The objectives established in [14] are closer to the objectives of our work. The paper describes a technique for robot navigation in a dynamic environment where pedestrians are moving. In [15], a dynamic window approach was combined with a deep RL technique, defining a reward function that reinforces velocities to move away from the obstacles. The work reported in [16] developed a system for navigation in a scenario with moving pedestrians detected by a vision system. A decentralized recurrent neural network is used to reason about the spatial and temporal relationships.

Most of the works using reinforcement and deep RL techniques take the perceived information directly from the sensors to compute the actions every sampling time. In the work presented in this paper, we use pre-processed environment information as input to the learning system adapted to the problem to be solved, in this case, autonomous navigation in dynamic environments. This approach benefits the training process and the performance of the system. In Section 7, a more detailed comparison of our work with other state-of-the-art works is developed.

## 3. DOVS Dynamic Environment Model

In this section, we summarize the main foundations and properties of the DOVS model [1]. It contains the input information to the RL system.

The model computes safe and unsafe velocities for a differential-drive robot and represents them on the velocity space. Velocities are calculated from the time to collision with the obstacles in a robocentric reference system. Maximum and minimum velocities and the corresponding times are computed from the intersection between a set of robot-feasible trajectories (compatible with the kinematic constraints) and the collision band swept by the moving obstacles.

Figure 1a represents a scenario with one robot and several moving obstacles in the workspace. Figure 1b shows the dynamic obstacle velocity–time space (DOVTS), in which the two 3D surfaces correspond to the two closest moving obstacles in this space. The time coordinates on this surface for each velocity (ω,v) represent their time to collision with the corresponding obstacle. Figure 1c depicts the dynamic obstacle velocity space (DOVS). The dark blue areas on the (ω,v) plane are the projection of the 3D surfaces on the velocity plane. These areas, which we name dynamic obstacle velocity (DOV), correspond to unsafe robot velocities that can lead to a collision if they are maintained over time. The white zones in this velocity space are safe velocities that generate collision-free trajectories. The DOVS contains the information that is used as input for the learning system. From it, the system selects an action (velocity) every sampling time among the safe velocities represented in the model. The centre of the green rhombus is the current robot velocity, and its contour depicts the maximum velocities that the robot can reach from the current velocity in the next control period, according to the acceleration restrictions. This way, the dynamic constraints are explicitly represented in the model. The straight blue line is the projection on the velocity plane of a circular trajectory to the goal with radius=v/ω.

Based on the DOVS model, in [1], the authors developed a set of navigation strategies associated with one of the situations identified in the model, the S-DOVS method. Each situation represents the relative locations and velocities between the robot and the obstacles, essential information for planning optimal actions for the next control periods. The action rules defined for each strategy allow the robot to pass at high velocity in front of the obstacle, to slow down to pass behind the obstacle, or to escape from a dangerous situation when it is moving at an unsafe velocity due to new obstacles appearing in the scene. More details about the model and the strategies can be found in [1].

The S-DOVS technique is used as a baseline to compare the learning-based methods developed in this work. One of the objectives is to analyze whether an RL-based technique outperforms a rule-based technique, in which the behavior rules are predefined for each situation.

## 4. System Overview

Figure 2 shows the general architecture of the system for autonomous navigation. From the workspace, the system captures the dynamic environment information duruing every sampling time and computes the DOVS model, including the current state and the predicted motion of the obstacles. A state vector is computed from this model, using the Q-learning algorithm to calculate the optimal utility values for each state. The optimal action is obtained from the resulting Q-table. Finally, the linear and angular velocities (v,ω) are computed from the action.

## 5. Reinforcement-Learning-Based Method

### 5.1. Q-Learning Algorithm

The RL method developed in this work for robot planning and navigation in dynamic environments is based on a Q-learning algorithm [2]. The inputs are the kinematic and dynamic information in the DOVS model. The outputs are the actions from which the linear and angular velocities to be applied during every sampling time are computed. The inputs to RL are processed to obtain the states. The continuous state values are discretized before feeding the RL input; then, the Q-tables that represent their utilities are obtained.

The Q(s,a) values are calculated by applying the expression:(1)$$Q^{*}\left(s,a\right)=Q\left(s,a\right)+\alpha \left(\left[R\left(s\right)+\gamma maxQ\left(s^{\prime },a^{\prime }\right)\right]-Q\left(s,a\right)\right)$$
where *s* is the current state; *a* is the current action; $s^{\prime }$ and $a^{\prime }$ are the state and action selected for the next iteration, respectively; α is the learning rate; R(s) is the reward at the state ‘*s*’; and γ is the discount factor.

There can be active agents (robots) and passive agents (moving obstacles) in a scenario. An active agent can make decisions about maneuvering, and the passive agents move following different trajectories. A scenario has at least one active agent. An episode is an execution of a scenario; the episode ends if all the active agents reach their goal, if one or more of them collide, if an active agent is too far from its goal, or if too many iterations have occurred.

In each episode, the agent chooses an action from the initial state using an ϵ−greedy algorithm for exploring and exploiting. During the training phase, the ϵ parameter is adapted to initially extend the exploration to the whole velocity space whilst exploiting the learned actions once the system has explored a great part of this space.

Instead of directly using the kinematic and dynamic information of the agents in the scenario as most other works have done, the method proposed here processes this information using the DOVS explained in Section 3. The safe and unsafe velocity commands represented in the DOVS allow one to reduce the potential velocities to explore, only selecting the ‘best’ one from the safe commands.

In this approach, all the active agents reason using the same optimal policy learned in the training stage. Thus, it is very important that the training is performed in a great variety of scenario states, including the total number of agents, number of active and passive agents, location of goals, random trajectories for the passive agents, random initial locations for agents and random linear and angular velocities for passive agents.

The use of the DOVS model as input to RL speeds the training process, because the information managed by the learning system is focused on the problem to be solved. Instead of utilizing raw information from the sensors or only very simple past information as other works have done, our method works on predictions about the further evolution of the scenario that are implicit in the model, allowing us to make more informed decisions.

The states, actions and rewards are defined in the following subsections.

### 5.2. States

Navigation in dynamic environments needs to account for the relative locations and velocities between the active agents and the other obstacles, as well as the collision risk with close obstacles. Instead of using the raw information from onboard sensors, such as lidar, as an input for the training phase, the environment dynamics are obtained from the DOVS model that pre-processes the sensor data.

The first step is discretizing the continuous variables involved in the DOVS model. If there are too many states, the process of learning could be excessively time-consuming, but a sufficient number of states to represent the environment correctly without losing relevant information are needed. It is necessary to reach a discretization commitment.

A state *s* is defined as:(2)$$s=(d_{goal},\theta_{goal},v_{agent},\omega_{agent},col,\theta_{rel},v_{obs})$$

The variables respectively correspond to linear and angular distances to the assigned goal, linear and angular agent velocities, the danger of collision, the relative orientation with respect to the obstacle and the velocity of the obstacle. Table 1 shows the values for every state. $CObs_{t}$ represents the closest obstacle at time *t*, and col=No indicates that the state col has the value No; in other words, the agent is not in danger of collision.

### 5.3. Actions

An agent selects the linear and angular velocities (v,ω) for the next control period inside the dynamic window (the green rhombus in Figure 1c), defined from the maximum linear and angular accelerations for a differential-drive vehicle. When the direction of motion is far from the goal direction, the actions that an agent can select are the velocities given by the corners of the dynamic window. These actions correspond to applying the maximum linear angular acceleration. The agent can also maintain the same velocity. For safety reasons, only safe velocities can be selected from the set of DOV obstacles (see Figure 1c). If there are no safe velocities, the agent is permitted to choose velocities that modify the agent’s current velocity in such a way that a free velocity is reached as soon as possible.

If the current motion direction is close to the goal direction, the agent can apply velocities inside the dynamic window that take the agent directly to the goal.

The set of actions *A* is defined as:(3)A=(Up,Down,Right,Left,GoalUp,GoalDown,GoalEqual)

The actions are represented in Figure 3.

### 5.4. Rewards

The goal of the algorithm is to maximize an accumulated reward. The agent is rewarded when it chooses a safe action that moves it towards the goal and penalized when the action chosen distances the agent from the goal or endangers the agent. There are two types of rewards: those applied when the agent takes an action and those given when the agent enters a terminal state.

Table 2 represents the reward terms and their values, where velGoal indicates that a velocity leading directly to the goal is chosen. An active agent is rewarded when it reduces the distance $\left(r_{d_{goal}}\right)$ or orientation $\left(r_{\theta_{goal}}\right)$ to the goal and for distancing from the closest obstacle, increasing the safety $\left(r_{obs}\right)$. Velocities that create smooth trajectories or decelerate the robot when it is near the goal are also positively rewarded and penalized otherwise $\left(r_{vel}\right)$.

Choosing actions that lead directly to the goal is rewarded to obtain natural trajectories. Changes in velocity are penalized to avoid constant switching between velocities $\left(r_{acc}\right)$. The agent is only rewarded for distancing itself from the closest obstacle $\left(r_{obs}\right)$ if the agent is in a state of collision.

In Table 2, $r_{action}$ represents the reward finally applied in a state. The $r_{action}$ reward provides positive or negative values depending on the result of the action. When the agent enters a terminal state, it is greatly penalized if it collides with an obstacle, if it travels far from the goal, or if it exceeds the maximum number of iterations. The agent receives a large reward when it takes actions leading to a stop on the goal. Otherwise, reward *R* (Equation (Equation 4)) is applied:(4)$$R=0.75r_{d_{goal}}+2r_{\theta_{goal}}+4r_{obs}+4r_{vel}+r_{acc}$$

### 5.5. New RL-DOVS Techniques

Based on the Q-learning algorithm, the state, the actions and the rewards defined, a Q-table including Q-values for the discrete state was built. We developed two methods:RL-DOVS-A: In this method, the system learns the actions described in Section 5.3 having the maximum utility in each exploitation step, that is, the maximum Q-value (Section 5.1) for each state;RL-DOVS-D: In this method, a driver subjectively chooses these actions for the current state in each iteration, instead of using the Q-values as in RL-DOVS-A.

Table 3 shows the differences in both methods regarding training and execution. The objective of developing both methods is to compare their results and understand how the automatically learned actions resemble those manually chosen by a driver in the loop in similar situations. We examine whether including a human in the control loop improves the behavior of the automatically learned system. Some of the required metrics are defined in the next section.

## 6. Experiments and Result Evaluation

In this section, the performance of three navigation algorithms is analyzed and compared: (i) S-DOVS, (ii) RL-DOVS-A and (iii) RL-DOVS-D. The first is the method developed in [1], based on the DOVS model summarized in Section 3. The other two are the RL methods developed in this work, RL-DOVS-A and RL-DOVS-D. The latter is a much more tedious training task.

The dimensions of the environment for training and evaluation were 20 m × 20 m. In each training episode, a new scenario was randomly created, with 20 passive agents and 1 active agent. All the velocities and starting positions of the agents were also random. In 20% of these scenarios, all the passive agents had null linear velocity, simulating a scenario with static agents. The density of moving obstacles in each scenario was maintained during the simulations.

For the system evaluation, four different types of scenarios were created, having different number of active and passive agents, and random velocities and starting positions. For each of the four types, 100 scenarios were randomly created, but they were the same for each of the three methods, in order to achieve a realistic comparison of the three methods.

### 6.1. RL-DOVS-A Training

The RL-DOVS-A navigation system was trained during 2500 episodes with a learning factor α=0.05 and a discount factor γ=0.5. Given the variability of the environment, a constant learning rate factor was applied [17]. A reduced learning factor was chosen to slowly update the Q-table over time to ensure convergence.

Different discount factor (γ) values were tested; the results can be seen in Figure 4. We conclude that using from intermediate to high values, we obtained good results in terms of success rate and average iterations.

Initially, exploration rate ϵ = 1, and it was reduced by 0.0005 with each episode. This way, after 2000 episodes, no more states were explored, and the agent only exploited. Optimized learning for different scenarios was considered, obtaining different Q-tables. However, only one Q-table was trained and later applied to all types of scenarios for the following reasons:States related to mobile and static obstacles are distinguished, so there is no advantage in training two separate Q-tables for each type of obstacle;A similar success rate can be obtained in training scenarios with 20 agents and scenarios with 10 agents if more training time is applied in the latter case;Given the variability and unpredictability of possible scenarios, it is more convenient to apply only one Q-table learned for a wide variety of scenarios than to have several specialized Q-tables.

The training took approximately 595,000 iterations for 2500 episodes. In each episode, a new scenario was randomly created and launched, with 20 passive agents and 1 active agent with random starting positions, goals and velocities. In 20% of these scenarios, the passive agents were static. The total number of states after the training was 8329 states, as seen in Figure 5. The total number of states incremented rapidly at the beginning and then more slowly until converging in a total number of states. This was expected because most of the states had already been explored.

The utility of a state is given by the highest value of the Q-values for a state, which corresponds to an action. In Figure 6, the utility of the states was initially low, but at the end of the training, the values of a few states increased substantially. This is because a policy was chosen, and the high-value states were continuously repeated. These increments were produced in the 1000-episode area, which was expected because the agent began to exploit more than explore.

Table 4 describes the 10 states with the highest utility U(s) once the system was trained. The first column represents the states, and the other columns represent the values of each, from the highest to the lowest values. From this table, we can conclude that the states with higher utility value were the states where the agent was on the objective, had a linear velocity close to null, was aligned with its goal and was in a no-collision situation. $S_{1}$ was the state with the highest value for the system. This was expected because braking in these states would have ended the episode, and the action taken would have received a great reward.

### 6.2. RL-DOVS-D Training

The training of the system RL-DOVS-D was performed over 200 complete episodes, changing between scenarios with 1 active agent and 20 passive moving and static agents, as was done with the RL-DOVS-A system. In each iteration of the scenarios, the human driver chose the best action from their perspective and intuition. In this option, the Q-table was completed without needing an exploration phase. The training duration was approximately two hours.

The criteria used by the driver to choose the actions are the following:If the agent is unaligned with the goal, apply maximum angular velocities until the agent is aligned;If the agent is aligned with the goal, increase the linear velocity until an elevated linear velocity is reached;If the agent encounters an obstacle, choose to pass before or after the obstacle, following the driver’s subjective criteria.

The different scenarios were trained with the same Q-table used with the RL-DOVS-A system. Figure 7 shows the evolution of the utility of the states. In this case, the utilities of some states increased from the start of the training because it was the trainer who chose the actions, and there was no exploration phase.

The states with the highest utility are shown in Table 5. The state of the highest utility was the same as that in the RL-DOVS-A system. In this case, there were states where the agent was far from the goal. This is due to a much shorter training time than was required by the RL-DOVS-A system.

### 6.3. Evaluation and Comparison of Systems

Once the two systems were trained, some tests were run to compare the results with the S-DOVS system. The following scenarios were simulated:One active agent and ten passive mobile agents;One active agent and twenty passive mobile agents;One active agent and twenty passive static agents;Three active agents and eighteen passive mobile agents.

For each type of scenario, 100 completely random scenarios were created, with random velocities and agent starting positions. The scenarios were the same for the different systems to obtain a realistic comparison of the results. The following metrics are defined:*Success rate*: the percentage of scenarios tested in which the agent reached its goal;*Iterations*: average number of iterations;$d_{obs}$: average distance to the closest obstacle;*v*: average linear velocity;$d_{goal}$: average distance to the goal;$\theta_{goal}$: average alignment with the goal.

Table 6 (a–d) shows the results for four scenarios. The mean values and the standard deviation for the last four metrics are represented. Only values where the scenario was a success were considered for these metrics. The best values for each metric is highlighted.

With a low number of passive agents, the success rate was high, but the systems with RL ended up being safer, with a higher success rate and larger average distance to the closest obstacle. The RL-DOVS-A system obtained the shortest times to goal, as seen in the number of iterations, because the RL-DOVS-A typically maintained higher linear velocities. The RL-DOVS-D also applied high linear velocities but obtained the longest times to goal, so we can conclude that it realized longer trajectories. The original strategy-based method was more conservative with respect to the maximum velocities applied, maintaining a slightly lower Success rate than RL-DOVS-A. Table 6a shows the results.

The systems with RL presented a higher standard deviation in the number of iterations than S-DOVS, so this technique was more consistent with the length of trajectories it made. This difference in standard deviations repeated itself during all the tests. The average distance to the closest obstacle was similar in the different systems, but RL-DOVS-D maintained more distance from the closest obstacle, because the driver was more conservative about moving away from obstacles.

However, the average distance from and alignment with the goal tended to be larger in the systems with RL. RL-DOVS-A tended to move further away from the goal, and RL-DOVS-D tended to deviate from the goal more. This outcome is compatible with the highest maximum velocities for these systems whilst moving away from the obstacles, leading to longer trajectories with more curvature. Table 6b shows the results with 1 active agent and 20 passive mobile agents.

As expected, adding more passive agents reduced the success rate in general, slightly incremented the average iterations and reduced the average distances to the nearest obstacle. With this said, the success rate of the RL-DOVS-A system was higher, except for the less-dense scenario, which means it is a safer system. The average distance to the closest obstacle of the RL-DOVS-D was the highest, but this system had the lowest success rate.

Adding more passive agents increased the average iterations in the RL-DOVS-A system, making that count larger than in the original system, even though the average linear velocity was still greater. The RL-DOVS-D system continued to be the worst system with respect to iterations. As in the first test run, the distance and deviation with respect to the goal were much higher in the trained systems. The results with 1 active agent and 20 static passive agents are shown in Table 6c. The average velocity was similar for both RL-DOVSs and higher than in S-DOVS. The highest number of iterations in RL-DOVS-D means that the trajectories are longer for this technique in general.

The results are similar to those of the previous scenario type, but in this case, the success ratio of RL-DOVS-A was the same as that of the S-DOVS system. A problem that the new systems have is that they are restricted to only apply safe velocities, that is, the ones out of the DOVS, unless all velocities are unsafe, so they can become blocked if the obstacles do not move. In the case of the S-DOVS system, the selection of temporally unsafe velocities inside dynamic obstacles is allowed in order to maneuver to reach safer velocities.

The results with 3 active agents and 18 moving passive agents are represented in Table 6d. After adding three active agents, the success rate was greatly reduced for all systems, which was expected, because a scenario is only considered successful if all active agents reach their goal. However, the comparison of results is similar to the scenarios with 20 passive mobile agents and 1 active agent, whereby the success rate and the average distance to the closest obstacle were greater for RL-DOVS-A, but the average iterations were lower in the S-DOVS system.

To summarize, the RL systems always keep higher average velocities than the strategy-based ones but achieve longer trajectories and time to goal in dense scenarios due to the fact that they execute safer trajectories farther from the obstacles. RL-DOVS-A kept average velocities 70% higher than S-DOVS, obtaining a time to goal only 13% greater due to longer trajectories. There were no large standard deviations with respect to the mean values of velocities and distance to the obstacles for the three methods.

The RL-DOVS-A technique exhibited the best success ratio with respect to the other two techniques. Although in RL-DOVS-D the actions selected by the driver led to theoretically safer trajectories farther from the obstacles than RL-DOVS-A, its success ratio was lower. It is likely that it is due to the more-reduced training time devoted to the RL-DOVS-D technique.

Figure 8 shows the scenarios at a given time step for the three methods. The execution of the three methods can be seen in the following videos: S-DOVS (https://youtu.be/kBxtYju9Qug; accessed on 6 April 2022), RL-DOVS-A (https://youtu.be/fIrwtF95v0Y; accessed on 6 April 2022) and RL-DOVS-D (https://youtu.be/irlmyGbREaM; accessed on 6 April 2022). The three methods were applied in scenarios with up to 10 and up to 20 moving and static passive agents with 1 active agent and others with 3 active agents.

One and three active agents were represented in these scenarios. In the active multi-agent scenario, the same technique was applied to all the agents, and the navigation method also worked correctly in this case. Each active agent considered the other agents as an obstacle; they did not need an explicit cooperation strategy to reach their goals.

The videos show that, in general, for the same scenario, the systems based on RL tended to create more circular and larger trajectories, whereas the strategy-based system tended to navigate to the goal more directly. However, the new systems compensated for these long trajectories by applying high linear velocities. RL-DOVS-D took more iterations than the other two techniques due to longer trajectories. The video shows that the active agents were not able to slow down to stop at the goal in one case. It could be due to the driver not braking on time; the driver might brake too late during the training phase. Therefore, the robot must come back to reach the goal. It does not occur in RL-DOVS-A because the system automatically applies braking when needed.

In scenarios with few obstacles, the models with RL had more room to maneuver; thus, they arrived at the goal sooner. This can be checked by comparing the average iterations of the results with 10 and 20 obstacles.

In scenarios with a larger number of obstacles, the S-DOVS system took more risks. If obstacles were between the active agent and its objective, the agent tended to traverse the area with obstacles. The systems with RL are more conservative by design and try to circle around the area with obstacles, because unsafe velocities are forbidden. This way, the RL-DOVS-A system obtained a higher success rate and larger average distance to the closest obstacle. However, increasing the number of obstacles left less space for the systems with RL to maneuver, so they took longer to reach the goal.

The linear and angular velocity profiles for six different scenarios and for the three methods are shown in Figure 9. The figure shows that each method, in general, exhibited a similar velocity profile for the six scenarios. The linear velocity profiles show that the systems with RL tended to maintain elevated linear velocities for longer. The S-DOVS system initially strongly accelerated until it reached the maximum velocity; then, it slowly reduced its speed until it reached the goal. This is the defined strategy for deceleration when approaching the goal. RL-based systems learned the best acceleration–deceleration behavior, accelerating until reaching a very high linear velocity, maintaining it during a long period and, finally, slowing down at maximum deceleration in the case of RL-DOVS-A. This way, this system reached the goal faster than the others. It can be also observed that RL-DOVS-A reached the goals with less oscillations that the other two techniques in several of the scenarios. This suggests that this method is able to better plan the motion of the next steps, reducing the more reactive behavior of the other techniques.

The angular velocity profiles show that S-DOVS applied high angular accelerations to re-orient the agent towards the goal. RL-DOVS-A obtained a smoother motion, computing lower maximum angular velocities. RL-DOVS-D provoked a more oscillatory motion to avoid the obstacles and re-orient towards the objective. This shows a more reactive behavior of the driver compared with a more planned motion using the utilities learned by RL-DOVS-A.

In summary, RL-DOVS-A exhibited a better global performance than the other two techniques. The system automatically learns the actions to apply in every state, needing neither prefixed strategies based on rules as S-DOVS does nor a driver to learn the actions, which increases the length and difficulty of the training. Note that the RL-based techniques are more restrictive regarding the velocities permitted, always applying safe velocities in the free velocity space. If these restrictions were relaxed, greater maneuverability would be obtained, possibly improving the results with respect to the rule-based technique. This investigation is left for future work.

The tests were performed using an Intel i5-10300H (8) @ 4.500 GHz CPU with 8 GB of RAM. The new reinforcement learning systems were built on top of the already existing DOVS system, which was developed in C++. The Q-learning algorithm was implemented without the use of pre-existing machine learning libraries in order to adapt the algorithm to the already existing system, offering more flexibility and control over the process. The average iteration time was calculated over a set of 100 different scenarios with 1 active agent and 20 passive agents, obtaining 2150 ± 472 μs for S-DOVS, 3617 ± 1338 μs for RL-DOVS-A and 3951 ± 1360 μs for RL-DOVS-D. A positive aspect of the developed Q-learning algorithm is that, once the system is trained, the agents only need to look up the best action to take in the Q-table depending on their current state. Even though the computation times are slightly higher for the reinforcement learning systems than for the strategy-based original method, they are small enough to be suitable for real-time applications. A higher standard deviation is expected, since computation time can vary depending on where the sought state is positioned on the Q-table.

## 7. Discussion

We compare, in this section, some results of the RL-DOVS-A method developed in this work with the best results of other methods in similar scenarios. Although the scenarios and experiments are not equal, we consider that the metrics used serve for comparing the performance of the methods. Table 7 presents these metrics. The system developed in [14] describes a technique (WALLS-I) for robot navigation in a dynamic environment where pedestrians are moving. The reward includes several terms related to the goal, obstacles, waypoints and task time, and the actions are the linear and angular velocities. The value function and policy are computed by an Actor-Critic algorithm, mapping from observation space to action space. No information about future predictions is achieved; only the current information is used to compute the policy.

In [15], a dynamic window approach is combined with a deep RL technique for navigation in dynamic scenarios using a rangefinder sensor. The approach combines the benefits of the dynamic window approach (DWA) in terms of satisfying the robot’s dynamics constraints with state-of-the-art deep-reinforcement-learning-based navigation methods that can handle moving obstacles and pedestrians. The method achieves these goals by embedding the environmental obstacles’ motions in a novel low-dimensional observation space. It also uses a novel reward function to positively reinforce velocities that move the robot away from the obstacle’s heading direction leading to a low number of collisions. The reward function reinforces velocities moving away from the obstacles. The method DWA-RL uses the history to compute the motion command. This technique is more conservative than ours because it reinforces velocities that locate the robot behind the obstacle but does not favor velocities that force the robot to pass in front of the obstacle by increasing the velocity and reducing the time to goal if possible. Our method predicts the motion and potential future collisions in a spatial-time horizon to compute a safe command, favoring the high linear velocities. It predicts the future motion of the obstacles, an a priori better solution than using only past obstacle movements.

The work reported in [16] developed a system for navigation, a decentralized–structural recurrent neural network (DS-RNN), in a scenario with moving pedestrians detected by a vision system (DS-RNN). A decentralized recurrent neural network is used to reason about the spatial and temporal relationships for robot decision making in crowd navigation. The system is trained with model-free deep reinforcement learning without any expert supervision. The scenario is represented as a decentralized spatio-temporal graph, with a set of nodes representing the agents, a set of spatial edges connecting agents at the same time step and the temporal edges connecting the same nodes at adjacent time steps. This system captures the interactions between the robot and multiple humans through both space and time. The system was tested for 90, 180 and 360 field-of-view (FOV) degrees, obtaining the best performance for 360 FOV. The use of the spatio-temporal reasoning improves other RNN-based systems that do not use this kind of reasoning. The method does not use the pedestrian velocities as ours does but instead only uses the current and past states. This is an end-to-end method that jointly learns the interactions and the decision making. The reward function is very simple, rewarding the goal proximity and penalizing the proximity to obstacles and collisions. In an open scenario, only five moving humans appear, not providing a success rate and, sometimes, performing long trajectories.

The results obtained in our work, analyzed in Section 6, outperform the ones reported in the previous works, as Table 7 shows. Note that our method explicitly considers the kinodynamic robot constraints to select every control period. Among the other works used for comparison, only DWA-RL takes these restrictions into account. Many works in this field do not apply the robot constraints. Training systems that avoid them leads to situations in which a high acceleration or deceleration must be applied to escape from a collision when using real robots. Therefore, if this information is not included during the training, the robot collides, and the collision may not be reflected in the simulation results.

The term ‘training environments’ refers to the different spaces used for training in each method. For each environment, multiple scenarios were created in which the initial and goal locations, obstacle velocities and the number of static and moving obstacles were randomly generated. Moreover, all the techniques tested the algorithms in multiple random scenarios that include static and moving obstacles. Our method exhibited similar results in scenarios with moving and static obstacles. It obtained a higher success rate in scenarios with a similar number of moving obstacles than other methods. In the case of the DS-RNN method, the success rate was higher, but there were only 4 moving obstacles, and the other 16 were static. Moreover, our method was trained with many fewer steps. The average velocity for a higher number of moving obstacles was greater in our method than that reported for the DWA-RL technique.

The three techniques with which we compared ours are based on deep and reinforcement learning, in which the policy is computed by the network. Our technique is Q-table-based reinforcement learning (Q-RL), differing from other classical Q-RL mainly in the way of computing the rewards for each state, which is performed by a model that provides elaborated information about the dynamism of the scene. As discussed in this section, from the results in Table 7, our system obtained better metrics in terms of success rate, training iterations and average velocities against the others and in bigger and denser scenarios having more moving obstacles. Apart from better performance results, the implementation of our RL-DOVS systems is much simpler than the deep-learning-based architectures; therefore, it is possible for it to be executed in real time by a low-cost computer. The number of parameters to be tuned for training is not very high, against the number of parameter and hyperparameters needed for a deep learning architecture.

## 8. Conclusions and Future Work

In this work, we developed two reinforcement-based techniques for robot navigation in dynamic environments. In RL-DOVS-A, the system automatically learns the best policy to apply in every control period from the state utilities. In RL-DOVS-D, the actions are learned from a driver who selects the subjective best velocity command for each control period from the state information of the scenario. This dynamic information is represented in the DOVS model, obtained from an onboard lidar sensor and used as the inputs for the learning system. One of the lessons learned from this work is that the system that automatically learns could improve the S-DOVS using pre-established fixed rules for the strategies.

From the evaluation in random scenarios, using different numbers of active and passive agents with random motion directions and velocities, we conclude that the RL-DOVS-A system works, on average, better than the original S-DOVS method and the RL-DOVS-D learning-based method trained by a human. The mean velocities are higher, and the success rate is better than those of the other systems. The RL-DOVS-A system might be improved if the maneuverability against moving obstacles were increased by allowing it to select unsafe velocities when needed or if a longer training time was used. However, this would be a tedious task for the trainer, especially for the RL-DOVS-D method, and not much more time should be spent in the training phase.

The resulting navigation systems with RL are more conservative with respect to the distance to obstacles compared with the original system, creating more circular and longer trajectories with higher linear velocities. As a result, on occasion, the trajectories are quicker with RL in less-obstacle-dense scenarios. As more obstacles are added, the agents have less space to maneuver, resulting in slower trajectories than the original system and fewer collisions.

As learned from the exhaustive experimentation in autonomous navigation in dynamic environments, many variables of the workspace and of the velocity space must be used to obtain optimal or suboptimal policies. This leads to the discretization of all these variables to generate manageable Q-tables. It is necessary to keep a balance between the number of states to correctly represent the environment and the generalization capability of the learning methods. The intuitive and empirical process applied to select the final values for the states and rewards, although partially derived from the rules in S-DOVS, was the result of the large number of decision variables needed and the many situations that appear in these complex scenarios.

In complex environments, it would be convenient to have a much higher number of states to represent the environment more precisely, even though it would mean creating a more complex and difficult problem. Greater resources with more processing power and memory could accomplish this task. However, other options must be explored. As in the referenced works, Q-learning and deep-Q-learning algorithms allow one to use a higher number of states or continuous state and action spaces to compute the optimal policies. In an ongoing work, we are developing a learning system based on these ideas to substitute the Q-tables with continuous utility functions. We claim that instead of applying as input the raw information from the onboard sensors (cameras, lidar), using the pre-processed information from the scene as we show in the current work contributes to making the training procedure easier and quicker in complex and rapidly changing or dynamic scenarios. Thus, we aim to apply the DOVS model to represent the dynamism in the new systems. More complex scenarios with many active and passive moving and static obstacles are to be evaluated.

## Figures and Tables

**Figure 1 sensors-22-03847-f001:**
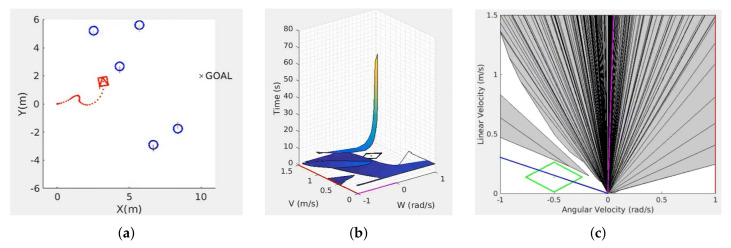
(**a**) Workspace with one robot (red) and several moving obstacles (blue); (**b**) velocity–time space *DOVTS* and velocity–time obstacle *DOVT* for two moving obstacles close to the robot and the corresponding projection on plane (ω,v), which is the dynamic obstacle velocity DOVS; (**c**) DOVS, in which the dark areas represent unsafe velocities, and white areas represent velocities that can be selected for safe navigation.

**Figure 2 sensors-22-03847-f002:**
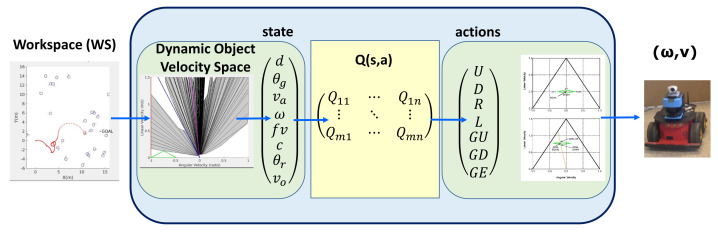
General architecture of the navigation system. WS represents the workspace; *Q*(*s*, *a*) is the Q−table, and (ω, *v*) are the velocities computed.

**Figure 3 sensors-22-03847-f003:**
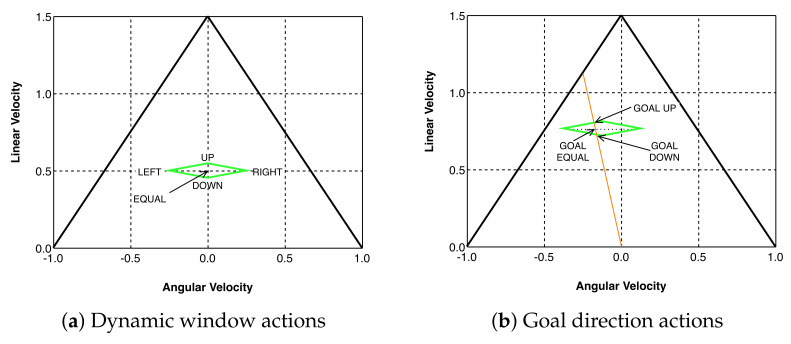
Actions: (**a**) the robot orientation is far from the goal direction; (**b**) the robot is near to being aligned with the goal.

**Figure 4 sensors-22-03847-f004:**
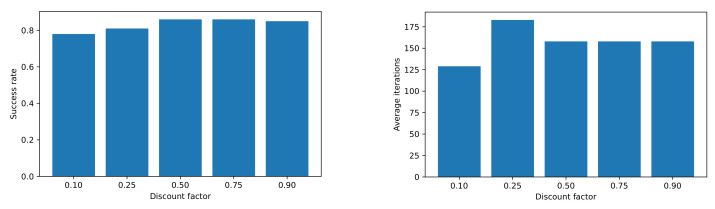
Results with different values for γ.

**Figure 5 sensors-22-03847-f005:**
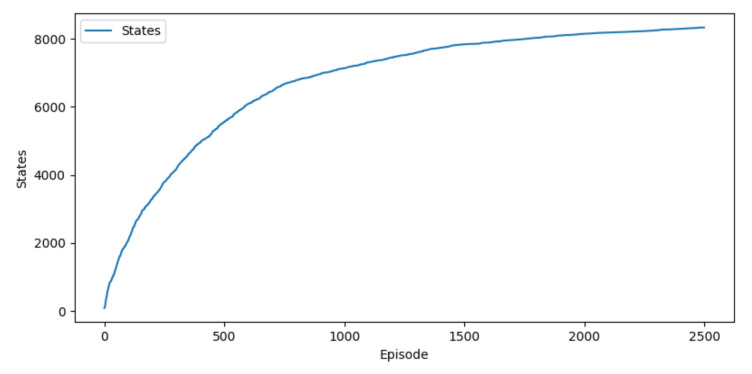
Total number of states during training.

**Figure 6 sensors-22-03847-f006:**
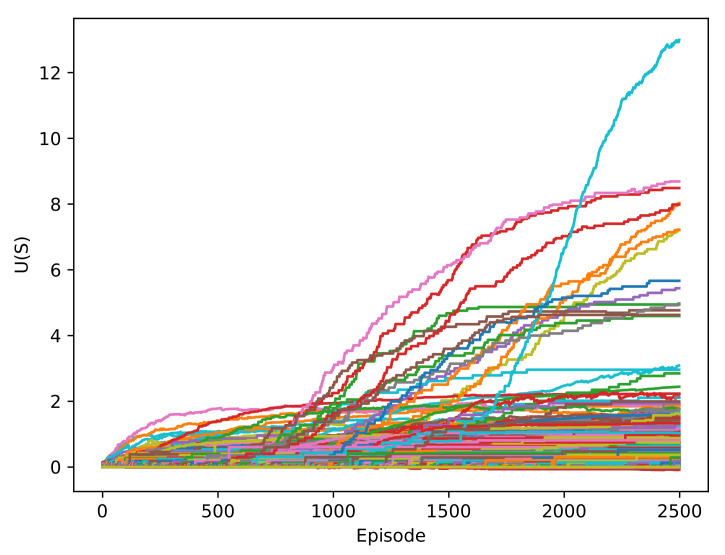
Evolution of the utility for all the states in the RL-DOVS-A system.

**Figure 7 sensors-22-03847-f007:**
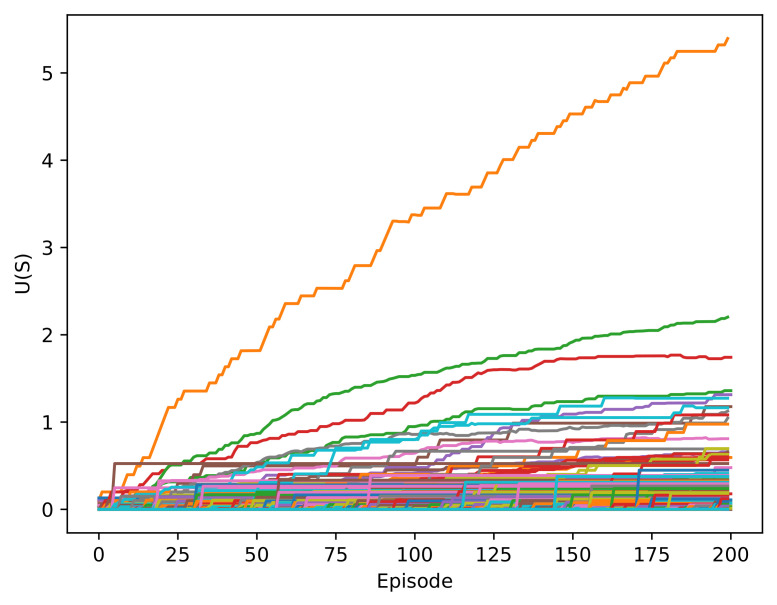
Evolution of the utility of all the states in the RL-DOVS-D system.

**Figure 8 sensors-22-03847-f008:**
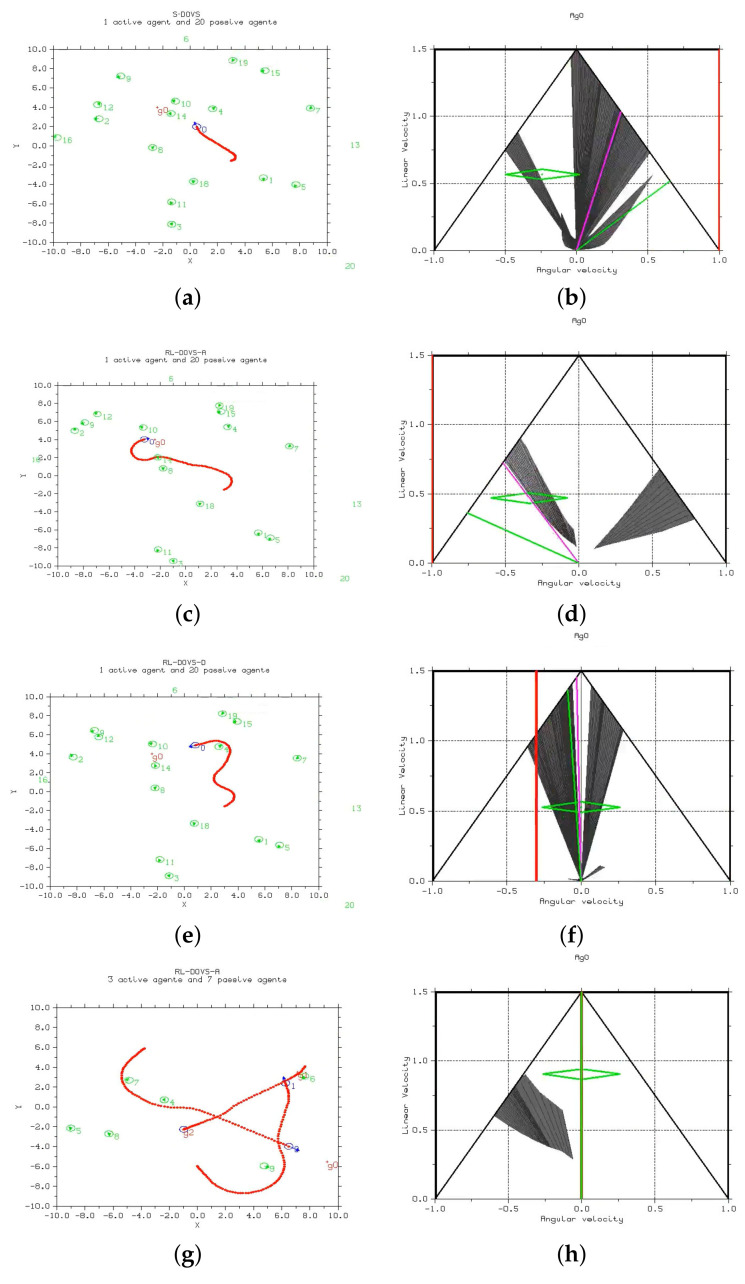
Workspace (**left**) and velocity space (**right**) for the three techniques and a given time step. In the workspace, the moving obstacles are in green, the robot trajectory in red. The dark areas correspond to the unsafe robot velocities that lead to collision with the surrounding obstacles. The dynamic window at the currect instant is depicted in green. (**a**) S-DOVS, 1 active-20 moving passive agents; (**b**) velocity space for S-DOVS; (**c**) RL-DOVS-A, 1 active and 20 moving passive agents; (**d**) velocity space for RL-DOVS-A; (**e**) RL-DOVS-D, 1 active and 20 moving passive agents; (**f**) velocity space for RL-DOVS-D; (**g**) RL-DOVS-A, 3 active and 7 moving passive agents; (**h**) velocity space for agent 0.

**Figure 9 sensors-22-03847-f009:**
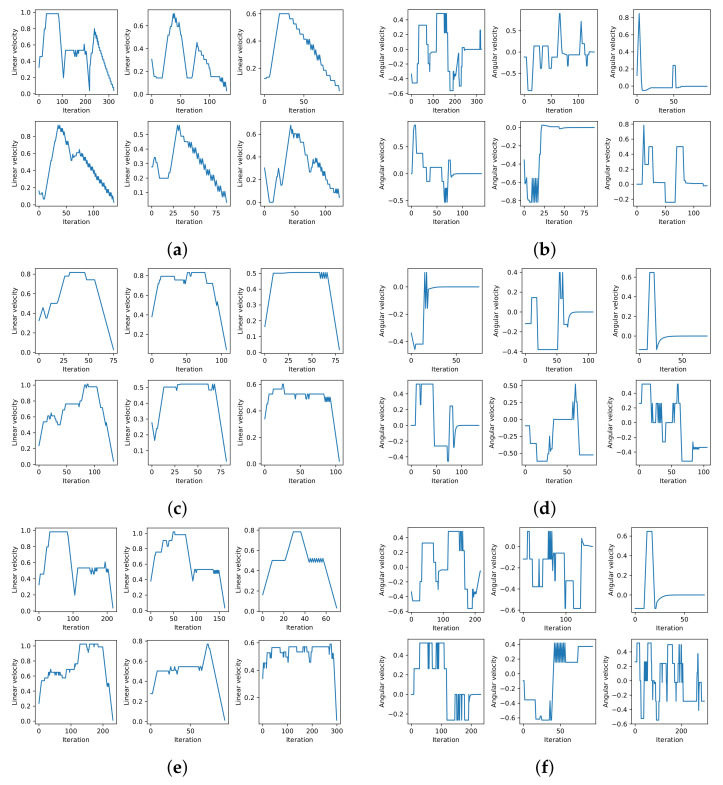
Velocity profiles for S-DOVS, RL-DOVS-A and RL-DOVS-D systems in six different scenarios. (**a**) Linear velocity S-DOVS. (**b**) Angular velocity S-DOVS. (**c**) Linear velocity RL-DOVS-A. (**d**) Angular velocity RL-DOVS-A. (**e**) Linear velocity RL-DOVS-D. (**f**) Angular velocity RL-DOVS-D.

**Table 1 sensors-22-03847-t001:** Discrete states for RL.

**State**	$d_{goal}$	range	$\theta_{goal}$	range	$v_{agent}$	range	$\omega_{agent}$	range
**variables**	On goal	<1	Aligned	[−0.1,0.1]	Very high	>1	Hard left	>0.5
	Very close	<2	Behind left	>$\frac{2\pi }{3}$	High	>0.75	Left	>0.1
	Close	<4	Behind right	<$-\frac{2\pi }{3}$	Medium	>0.5	Straight	[−0.1,0.1]
	Far	<8	Left	>0	Low	>0.1	Right	>−0.5
	Very far	<40	Right	otherwise	Very Low	otherwise	Hard right	otherwise
	Too far	≥40						
**State**	col	range	$\theta_{rel}$	range	$v_{obs}$	range		
**variables**	Left	$d_{obs}<2$∧	NoCollision	col=No	NoCollision	col=No		
		$\theta_{obs}\in \left[0,\frac{\pi }{3}\right]$	ToLeft	>$\frac{2\pi }{3}$	High	$v_{obs}>1$		
	Right	$d_{obs}<2$∧	MPerpLeft	>$\frac{\pi }{3}$	Medium	$v_{obs}>0.5$		
		$\theta_{obs}\in \left[-\frac{\pi }{3},0\right)$	SameLeft	>0	Low	$v_{obs}>0$		
	No	$d_{obs}\geq 2$∨	SameRight	>$-\frac{\pi }{3}$	Fixed	otherwise		
		$CObs_{t}\neq CObs_{t-1}$	PerpRight	>$-\frac{2\pi }{3}$				
			ToRight	otherwise				

**Table 2 sensors-22-03847-t002:** Rewards.

**Reward**	$r_{d_{goal}}$	condition	$r_{\theta_{goal}}$	condition	$r_{obs}$	condition
**variables**	$-\Delta d_{goal}$	$d_{goal}>1$	$-\Delta \theta_{goal}+0.3$	$d_{goal}>1$∧velGoal	$\theta_{obs}-\theta_{obs_{int}}-0.3$	col≠No
	0	otherwise		$\wedge \theta_{goal}\in \left(-\frac{\pi }{3},\frac{\pi }{3}\right)$	0	otherwise
			$-\Delta \theta_{goal}$	$d_{goal}>1$		
			0	otherwise		
**Reward**	$r_{vel}$	condition	$r_{acc}$	condition	$r_{action}$	condition
**variables**	0	col≠No	−0.3	Δv≠0	−10	collision
		∨Δv=0	0	otherwise	−5	iter≥1000∨ $d_{goal}\geq 40$
	0.20	Δv>0.01			10	$d_{goal}\leq 0.5\wedge v=0$
	−0.20	Δv<0.01			*R* (Equation (Equation 4))	otherwise
	−0.1	otherwise				

**Table 3 sensors-22-03847-t003:** Differences between RL-DOVS-A and RL-DOVS-D.

		RL-DOVS-A	RL-DOVS-D	Scenarios
**Training**	Action	Depending on ϵ:	In each iteration, the	100
	selection		driver selects an action	random
		**Exploration**: the agent	independently	for each
		applies a random action	of the Q-table	of the
		**Exploitation**: the agent	observing the	2500
		selects the best action	workspace and the	episodes
		by looking up its	velocity space	
		current state in the Q-table		
	Q-table	The Q-table value for the state		
	update	is updated using Equation (Equation 1)		
**Execution**		The agent selects the best action		100
		by looking up the trained Q-table		random,
				the same
				for all
				the methods

**Table 4 sensors-22-03847-t004:** The 10 most valuable states after training the RL-DOVS-A system. V.Low represents the state Very low; B.Left and B.Right represent the states Behind left and Behind right respectively; NC represents the state No collision.

State	*v*	*w*	$d_{goal}$	$\theta_{goal}$	$\theta_{rel}$	$v_{obs}$	Collision
$S_{1}$	V.Low	Straight	Goal	Alligned	NC	NC	NC
$S_{2}$	V.Low	Right	Goal	B.Right	NC	NC	NC
$S_{3}$	V.Low	Left	Goal	B.Left	NC	NC	NC
$S_{4}$	V.Low	Right	Goal	Left	NC	NC	NC
$S_{5}$	V.Low	Straight	Goal	B.Right	NC	NC	NC
$S_{6}$	V.Low	Left	Goal	Right	NC	NC	NC
$S_{7}$	V.Low	Straight	Goal	B.Left	NC	NC	NC
$S_{8}$	V.Low	Left	Goal	B.Right	NC	NC	NC
$S_{9}$	V.Low	Right	Goal	B.Left	NC	NC	NC
$S_{10}$	V.Low	Left	Goal	Left	NC	NC	NC

**Table 5 sensors-22-03847-t005:** The 10 most valuable states after training the RL-DOVS-D system. V.Far represents the value Very far and N.C. is no collision.

Variable	*v*	*w*	$d_{goal}$	$\theta_{goal}$	$\theta_{rel}$	$v_{obs}$	Collision
$S_{1}$	V.Low	Straight	Goal	Alligned	N.C.	N.C.	N.C.
$S_{2}$	Low	Straight	Goal	Alligned	N.C.	N.C.	N.C.
$S_{3}$	Medium	Straight	V.Far	Alligned	N.C.	N.C.	N.C.
$S_{4}$	Low	Straight	V.Far	Alligned	N.C.	N.C.	N.C.
$S_{5}$	Low	Right	V.Far	Right	N.C.	N.C.	N.C.
$S_{6}$	V.Low	Straight	Goal	B.Right	N.C.	N.C.	N.C.
$S_{7}$	V.Low	Straight	Goal	Left	N.C.	N.C.	N.C.
$S_{8}$	Medium	Left	V.Far	Left	N.C.	N.C.	N.C.
$S_{9}$	V.Low	Straight	V.Close	Alligned	N.C.	N.C.	N.C.
$S_{10}$	V.Low	Straight	Goal	B.Left	N.C.	N.C.	N.C.

**Table 6 sensors-22-03847-t006:** Results with different numbers of active and passive agents.

(a) Results with 1 active agent and 10 mobile passive agents.			
Metric	S-DOVS	RL-DOVS-A	RL-DOVS-D
*Success rate*	0.94	0.96	**0.98**
*Iterations*	112.10±29.17	94.93±62.14	117.89±73.25
$d_{obs}$	3.83±1.51	3.85±1.52	4.01±1.42
*v*	0.34±0.10	0.56±0.13	0.59±0.14
$d_{goal}$	4.11±1.99	4.96±2.6	4.18±2.04
$\theta_{goal}$	0.22±0.17	0.48±0.37	0.72±0.49
(**b**) Results with 1 active agent and 20 mobile passive agents.			
Metric	S-DOVS	RL-DOVS-A	RL-DOVS-D
Success rate	0.84	**0.89**	0.81
Iterations	123.84±50.45	130.05±75.34	156.46±85.37
$d_{obs}$	2.63±0.95	2.64±0.99	2.77±0.74
*v*	0.34±0.10	0.58±0.14	0.59±0.15
$d_{goal}$	4.3±2.21	5.86±2.98	4.69±2.23
$\theta_{goal}$	0.29±0.22	0.74±0.36	0.91±0.34
(**c**) Results with 1 active agent and 20 static agents.			
Metric	S-DOVS	RL-DOVS-A	RL-DOVS-D
Success rate	**0.92**	**0.92**	0.84
Iterations	112.15±34.54	128.29±82.89	152.38±101.19
$d_{obs}$	2.27±0.86	2.24±0.82	2.42±0.76
*v*	0.35±0.10	0.56±0.13	0.56±0.15
$d_{goal}$	4.22±2.07	5.09±2.64	4.27±2.18
$\theta_{goal}$	0.24±0.15	0.80±0.38	0.92±0.37
(**d**) Results with 3 active agents and 18 moving passive agents.			
Metric	S-DOVS	RL-DOVS-A	RL-DOVS-D
Success rate	0.56	**0.63**	0.29
Iterations	159.80±37.39	167.81±55.82	228.69±100.92
$d_{obs}$	2.36±0.81	2.64±1.01	2.95±0.862
*v*	0.27±0.10	0.43±0.20	0.43±0.21
$d_{goal}$	3.50±1.84	4.40±2.59	3.52±2.00
$\theta_{goal}$	0.27±0.21	0.77±0.47	0.89±0.56

**Table 7 sensors-22-03847-t007:** Comparison with the best results of other methods. In the training environments and test scenarios, the initial and goal locations, obstacle velocities and the number of static and moving agents were randomly generated.

Metric	DWA-RL	WALLS-I	DS-RNN	RL-DOVS-A
moving obstacles	17	16	5	20
trained agents	4	1	1	1
training environments	1	6	3	1
test scenarios	50	300	500	100
success rate	0.42	0.80	0.96	0.89
$v_{average}$	0.38	-	-	0.58
learning iterations	700 k	200 k	10,000 k	595 k

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
