# Peer review of "RL-DOVS: Reinforcement Learning for Autonomous Robot Navigation in Dynamic Environments"

_sensors, 2022, doi:10.3390/s22103847_

Round 1
Reviewer 1 Report
The authors have proposed a new robot path planner for navigation in dynamic environments. The proposed approach is based on reinforcement learning, and it uses robocentric dynamic environment model data as input for the learning process.
The paper is well constructed. The review of related research works includes the works from recent years, and it sufficiently covers the research area. On that basis, the authors have formulated the research problem and proposed a new algorithm for solving it. The two proposed variants of the reinforcement learning-based path planning algorithms were presented and compared experimentally with the original Dynamic Object Velocity Space-based path planning algorithm, which the authors have used as starting point for their research. The paper is interesting and proposes a new approach for robot trajectory planning in dynamic environments.
The following issues should be addressed before publication:
1. The quality of figure 8 should be improved. Charts are blurry.
2. The quality of the English language needs to be improved. There are a number of grammatical, punctuation and stylistic errors in the article.
3. As many types of scenarios were used and within each type of scenario 100 scenarios were generated randomly (as the authors claim), so the results (for example presented in tables 5 and 6) should be statistically analyzed. Now, it is not clear how the metric values are computed, whether the results are mean values and from which type of scenarios they come from.
Reviewer 2 Report
The authors present a case for reinforcement learning that takes into account new facets of environmental data that accommodates the dynamic data of the same. And, indeed, the model for actions for maximum utility and the model where the best driver actions are learned is an interesting concept for the field to consider.
There are a few areas that can help the reader better appreciate the research that has been conducted.
Figure 1 is presented very early in the paper and there is the expectation that this will be discussed, though it is much later in the paper that a meaningful exploration of the figure is actually discussed. Reconciling this will add to the flow of the paper.
Some form of tabular comparison of -A and -D systems would help the reader as the paper introduces these two new models but also is comparing them to the literature. This takes re-reading to follow what the key attributes of the new models are relative to the literature.
Figures 1 and 2 are informative, but they can be difficult to read in detail due to having to infer, say, what WS is (on Figure 2). Is there a way to more clearly represent the WS and "action" images?
Figure 4 seems lost in the paper and it is not clear if this is every discussed in the narrative or the significance addressed.
Figure 9 is potentially extended clarification of Table 5 and Figure 8, but the significance is lost since the images are so small. Does Figure 9 really help the case for comparing the performance differences between the three systems?
The discussion of the new dynamic systems appears to have improved performance, but there does not appear to be mentioned the computational cost. For real-time navigation, this issue is significant. If the results are computationally inefficient, then the improved performance may not have practical impact.
Computing platform and languages used do not seem to be mentioned, and by what mechanism (if at all) would the algorithms be available for others to explore. The technique appears to have promise with improved environmental sensing, but the paper relies on references for the reader to seek out to set a foundational understanding of the algorithm. This is not a bad thing, but makes a fuller understanding of the impact of the new algorithm more difficult to appreciate for those working in this field.
Reviewer 3 Report
- In abstract, the full name for “RL-DOVS” is required.
- In Figure 1 (b) and (c), clear explanations are needed, even though there are some comments in Section 3.
- Figure 2 has some redlines from inadequate captures.
- Also, Figure 2’s symbols have to be explained.
- Section 2 has to be reorganized with clear literature review. Currently, the detailed background and literature review lacks.
- Section 5.1 has to be moved to Section 2, as the provided are common.
- More explanation why the discrete states are provided has to be explained. Is there any continuous variable?
- GEq (Goal Equal) and similar terms have to be explained clearly.
- The dynamic situations and implementation have to be provided for Section 6.
- Section 6 fail completely to deliver how the dynamic environment are assumed and tested for the comparison results.
- In Section 6, explain the situations with Figure 8 and Figure 9 first.
- The compared methods (DWA-RL, DS-RNN)’s detailed features have to be provided.
- The advantages over the current deep q-learning using the framework have to be explained with the clear comparisons.
Round 2
Reviewer 1 Report
The authors have addressed the most important issues, so in my opinion the paper can be accepted.